# Machine Learning Applications in Optical Fiber Sensing: A Research Agenda

**DOI:** 10.3390/s24072200

**Published:** 2024-03-29

**Authors:** Erick Reyes-Vera, Alejandro Valencia-Arias, Vanessa García-Pineda, Edward Florencio Aurora-Vigo, Halyn Alvarez Vásquez, Gustavo Sánchez

**Affiliations:** 1Departamento de Electrónica y Telecomunicaciones, Instituto Tecnológico Metropolitano, Medellín 050013, Colombia; vanessagarciap@itm.edu.co; 2Escuela de Ingeniería Industrial, Universidad Señor de Sipán, Chiclayo 14001, Peru; 3Escuela Profesional de Ingeniería Agroindustrial y Comercio Exterior, Universidad Señor de Sipán, Chiclayo 14001, Peru; efaurora@uss.edu.pe; 4Facultad de Ingeniería, Arquitectura y Urbanismo, Universidad Señor de Sipán, Chiclayo 14001, Peru; avasquezh@uss.edu.pe; 5Instituto de Investigación y Estudios de la Mujer, Universidad Ricardo Palma, Lima 15074, Peru; sanchezsantosgustavo@gmail.com

**Keywords:** machine learning, fiber sensors, PRISMA, deep learning, fiber Bragg grating

## Abstract

The constant monitoring and control of various health, infrastructure, and natural factors have led to the design and development of technological devices in a wide range of fields. This has resulted in the creation of different types of sensors that can be used to monitor and control different environments, such as fire, water, temperature, and movement, among others. These sensors detect anomalies in the input data to the system, allowing alerts to be generated for early risk detection. The advancement of artificial intelligence has led to improved sensor systems and networks, resulting in devices with better performance and more precise results by incorporating various features. The aim of this work is to conduct a bibliometric analysis using the PRISMA 2020 set to identify research trends in the development of machine learning applications in fiber optic sensors. This methodology facilitates the analysis of a dataset comprised of documents obtained from Scopus and Web of Science databases. It enables the evaluation of both the quantity and quality of publications in the study area based on specific criteria, such as trends, key concepts, and advances in concepts over time. The study found that deep learning techniques and fiber Bragg gratings have been extensively researched in infrastructure, with a focus on using fiber optic sensors for structural health monitoring in future research. One of the main limitations is the lack of research on the use of novel materials, such as graphite, for designing fiber optic sensors. One of the main limitations is the lack of research on the use of novel materials, such as graphite, for designing fiber optic sensors. This presents an opportunity for future studies.

## 1. Introduction

The need for simpler monitoring and verification of different states and elements in nature has led to the creation of these sensors. The development of sensors of various types and materials has facilitated the monitoring of temperature, fire, water, air, and the detection of anomalies in these elements. Sensors have also been developed to monitor aspects such as glucose levels, sleep apnea, and heart rate, among others, in the field of health. Advancements in sensor development have rapidly progressed in various fields of knowledge where vibrations, movement, and other parameters are monitored. These sensors have become more adaptable to contextual needs, cost-effective, and faster. However, recent advances in hardware, such as optical fiber, and software, such as machine learning, have changed the configuration of these sensors. This allows them to more easily adapt to the specific needs of the context for which they are intended. As a result, various sensors have been developed for detecting and monitoring physical and sensory elements [1]. The emergence of various types of sensors has resulted in progress in detecting fire, water, temperature, and movement, among other things. These sensors identify anomalies in the input data received by the implemented systems [2]. Sensor design has evolved to include features that provide greater precision in operation [3]. Bioelectrical, biophysical, and biochemical signals can be employed in various contexts, including sports development, medical analysis and prevention, as well as rehabilitation [4]. Sensors play a crucial role in risk prevention, including health, road, natural, and architectural risks [5]. It is important to design, manufacture, and implement sensors in various contexts to mitigate these risks.

In recent years, new tools have been identified to facilitate the generation of detection systems [6]. Optical fiber is a widely used material in sensor design due to its unique characteristics and properties that facilitate signal manipulation. This results in the creation of more sensitive sensors with faster and cleaner signals [7]. However, fabricating fiber optic-based sensors can be a cumbersome process due to the integration of interferometers and the need for careful attention to amplitude modulation in the output spectrum [8]. Despite significant progress, challenges persist in the manufacturing and operation of these sensors, including complexity, associated costs, and technical limitations in design and manufacturing [9].

Artificial intelligence techniques can improve the efficiency and accuracy of sensors [10]. Current research focuses on applying various machine learning techniques to design more sensitive sensors and improve their performance [11]. This enables the classification and quantification of better samples [12]. Furthermore, machine learning techniques have been used to generate models that analyze the correlation between different parameters of the optical spectrum [13,14]. Fiber optic sensors have a wide range of applications, from industrial process monitoring to medical diagnosis [15]. A recent study proposed a novel method for assessing the health status of athletes in sports medicine using optical sensors and quantum computing. The data collected from optical sensors is analyzed using a recurrent neural network (ResNet) adapted for quantum convolutional learning (NResNetRNN_QCL) [4]. In this additional study, a machine learning (ML) approach is used to analyze musical signals in a health monitoring system, in conjunction with an optical sensor. The study applies quantum photonics and the optical sensor model to examine musical signals using a Markov propagation model with a reinforcement gradient vector to assess the observed data based on optical sensors (RGVMP) [16].

Significant progress has been made in the use of machine learning and optical sensors in various applications, including healthcare. For example, ref. [17] proposed an optical immunosensor that incorporates machine learning and signal transduction encoded with microspheres. This approach is used for the rapid and multiplexed detection of antibiotics in milk. The system uses polystyrene nanoparticles that are easily functionalized and vary in size and quantity. These nanoparticles carry multiple signal probes, allowing for the simultaneous detection of various antibiotics through a simple one-step signal conversion reaction. In another study, researchers employed a set of label-free colorimetric sensors based on Fabry–Pérot films with an organic-metallic structure to detect volatile organic compounds and food deterioration. Machine learning algorithms are used to analyze sensor data and accurately identify volatile organic compounds with a high classification accuracy of 92% at 100 ppm. This approach presents an economical platform of small colorimetric sensors that can be used to assess food quality, alterations, and deterioration. The platform has the potential to contribute to the development of smart labels and packaging [18].

Given the wide range of applications for this type of sensor and the variety of machine learning techniques that can be used in their design, it is necessary to synthesize and identify the main trends in the field to propose a future research agenda [19]. Despite significant advances in the design and development of technological devices for detecting and monitoring various physical and sensory elements, challenges and limitations still exist in existing approaches. One major challenge in manufacturing fiber-optic-based sensors is their complexity and cost. The process typically involves integrating interferometers and carefully managing amplitude modulation in the output spectrum, which can be cumbersome and laborious [20].

Despite the wide range of applications for optical sensors, their design and manufacture present specific technical difficulties. For instance, conventional methods such as Bragg gratings or Fabry–Perot etalons may have limitations in detecting specific targets due to the challenges of creating forward-looking probes (Bragg gratings) or achieving adequate signal-to-noise ratios (Fabry–Perot systems). These limitations can impact the accuracy and dependability of the data obtained, emphasizing the importance of addressing the technical challenges involved in the production and operation of optical sensors. Distributed fiber optic sensors (DFOS) have been widely used in infrastructure monitoring. The most commonly measured quantities are deformation (distributed deformation sensing—DSS), temperature (DTS), and acoustics/vibrations (DAS/DVS) [6]. However, DFOS can also directly or indirectly measure other types of quantities, such as displacement [21,22], pressure, force, relative humidity [23,24,25,26,27], radiation [28,29,30], gas concentration [31,32], and more.

The purpose of this study is to conduct a bibliometric analysis of the literature to identify research trends related to the development of machine learning applications in fiber optic sensors. Considering the constant growth of information and knowledge on the topic, it is necessary to evaluate the dissemination of knowledge on components related to the quantity and quality of research processes. This evaluation depends on the authors, countries, journals, and emerging concepts that will be crucial for the future development of the subject [33]. In order to achieve the study’s objective, the following research questions are proposed:RQ1: What is the historical evolution of scientific literature on machine learning applications in fiber optic sensors?RQ2: What are the primary research sources for machine learning applications in fiber optic sensors?RQ3: What is the thematic evolution of machine learning applications in fiber optic sensors?RQ4: What are the primary themes in using machine learning for fiber optic sensor applications?RQ5: What are the established and emerging keywords in the research field of machine learning applications in fiber optic sensors?RQ6: What topics are relevant for designing a research agenda on machine learning applications in fiber optic sensors?

For this purpose, the document is structured as follows; the methodology section explains how the literature review process is carried out and presents the research results. The text is free from grammatical errors, spelling mistakes, and punctuation errors. The following sections will discuss the results, propose a research agenda, and draw a conclusion on the most relevant elements of the research. The language used is clear, concise, and objective, with a formal register and precise word choice. The text adheres to conventional structure and formatting features and follows a logical progression with causal connections between statements. No changes in content have been made.

## 2. Materials and Methods

In accordance with the objective of the research and to answer the research questions, a bibliometric analysis is proposed that integrates the descriptive elements of the written information and is based on the documents; the results obtained will always be of a scientific nature [34]; this will increase the sources of detail and the reproducibility of the methodological model, which will be carried out in accordance with the PRISMA 2020 declaration, which defines the eligibility criteria, sources of information, search strategies, and management of the data obtained from the execution of this strategy. GIA in the selected data source [35].

### 2.1. Eligibility Criteria

To explore the applications of machine learning in the field of optical sensors, bibliometrics uses a set of inclusion criteria to guide the selection of articles for analysis. These criteria prioritize articles that have the combination of “machine learning” and “optical sensor” in their titles or keywords. This combination is considered basic metadata to identify the thematic relevance of each study. This selection strategy focuses on finding papers that specifically address the intersection between two cutting-edge technological areas. This promotes a detailed and specialized analysis of their evolution, trends, and contributions to the field.

The process of eliminating records consists of three phases designed to ensure the integrity and quality of the sample analyzed. In the first phase, records with indexing errors are excluded to eliminate potential biases or inconsistencies in the collected data. In the second phase, an additional exclusion criterion is applied specifically to documents without full-text access. It is important to note that this criterion is only applied to systematic literature reviews, as specified in [35], in order to maintain the coherence and reliability of the information available for this type of study.

It is worth mentioning that bibliometrics is limited to metadata analysis, as detailed in the reference. Finally, the third phase of exclusion focuses on the exclusion of conference proceedings due to their limited scope and methodological rigor compared to articles from scientific journals. This ensures the consistency and quality of the documents included in the bibliometric analysis. The rigorous methodological approach adopted in this study ensures the robustness and validity of the results obtained.

### 2.2. Information Sources

The sources of information for this bibliometric analysis correspond to scientific documents related to optical sensors and machine learning stored in the Web of Science and Scopus databases, which are the two most used databases for this type of analysis [36].

### 2.3. Search Strategy

To perform an efficient search in both databases, so that all the keywords defined as inclusion criteria can be included, two similar specialized search equations are designed, whose only variation is given by the search interface of each database. In this case, we have the following search equations: 

For the Scopus database: (TITLE (“machine learning”) AND TITLE (“optical sensor*”)) OR (TITLE (“machine learning” AND “optical sensor*”)) OR (KEY (“machine learning” AND “optical sensor*”)).

For the Web of Science database: (TI = (“machine learning”) AND TI = (“optic sensor*”)) OR (TI = (“machine learning” AND “optical sensor*”)) OR (TI = (“machine learning” AND “optical sensor*”)).

### 2.4. Data Management

After completing the search strategy, we obtained a total of 189 initial articles from the Scopus and Web of Science databases. Of these, 172 came from Scopus and 17 from Web of Science. These articles will be extensively analyzed using specialized IT tools, including Microsoft Excel and the open access software VOSviewer 1.6.19, which allows to apply bibliometric criteria that tend to qualify as measurable, objective, and multidimensional [37]. Specifically, we will use Microsoft Excel to perform a detailed analysis of bibliometric indicators, both in terms of quantity and quality. This tool allowed us to analyze various aspects, including the production of the main authors, the most influential journals, and the leading countries in research on the topic.

In addition, it helped us to identify the evolution of topics over time and to detect emerging words, which contributed to the development of a research agenda in the field of study. The VOSviewer 1.6.19 software was used to analyze the thematic co-occurrence in the recovered documents, identifying conceptual connection patterns and facilitating the visualization of knowledge networks in the scientific literature on the topic. This methodological approach allows a comprehensive and multidimensional understanding of the bibliometric panorama in the area of study, contributing to a precise and rigorous evaluation of the scientific production in this field.

### 2.5. Selection Process

The development of the method is carried out independently by each of the authors of the article, and the differences are addressed collectively. To summarize the process, Figure 1 shows the first phase of identification, the second of screening with the three defined exclusion phases, and finally, there are 126 documents that will be studied in the present bibliometric analysis.

## 3. Results

Taking into account the results obtained from the publications in the Scopus and WoS databases, in Figure 2, the growing interest in the topic can be observed, noting that the first publications related to machine learning and fiber optic sensors were carried out around 2012. However, from 2017 onwards, the research related to the topic was constant, and the interest showed a growing trend from 2018 onwards, showing a greater number of publications on the topic in 2022, with 40 publications related to machine learning and fiber optic sensors, in addition to 24 articles so far in 2023. The technical aspects and contributions of some of the most important works by time period are highlighted below. Among the most cited articles for the year 2018, in which significant interest in the topic began, there is the research of [38], which developed a system capable of classifying disturbances according to the location where they occurred along a multimode optical fiber, based on the training of a learning algorithm automatically and classifying subsequent disturbances based on the spatial locations where they were found.

Another representative article among the research related to machine learning and fiber optic sensors is the article by [39], who developed a vehicle classification system based on fiber optic Bragg grating sensors reinforced with glass fibers. (3-D FRP-FBG) using support vector machine learning algorithms to classify vehicles into categories ranging from small cars to combination trucks. By 2022, one of the most cited research papers proposes and tests deep learning models trained with real seismic data to detect earthquakes in fiber optic distributed acoustic sensor (DAS) measurements [40]. On the other hand, among the most recent researches, a sensor arrangement method based on the characteristics of the sensor and the fire field is proposed to better reflect the temperature changes of the fire in the building space and conduct high temperatures by establishing several prediction models using different algorithms based on the dataset and the models built by artificial neutral network (ANN) and long short-term memory (LSTM) to obtain better performance [3].

In terms of the most representative authors on the subject, Figure 3 shows the main references in the field of study of fiber optic sensors and machine learning, the figure shows that, the group in yellow corresponds to the authors with more publications and more citations, the group in blue corresponds to the authors with more citations, but no more publications, and the group in green corresponds to the most cited authors, but with fewer publications. Below are the technical aspects and contributions of some of the most relevant papers by author group. Among the main authors, we can find Frizera A. and Liu Y., who are the authors who have done the most research on the subject and who have the most cited publications. Among the most relevant works of these authors is research that presents the working principle and the main characteristics of the specklegram and analyzes in detail the applications of fiber specklegram sensors (FSS), also discussing the advances in microelectronics, machine learning, material processing for new optical fibers, and their relationship with FSS [1].

Then there are the authors who, although they do not have a large number of publications, their research is the most cited. Among these authors are: Sierra-Perez, J. and Alvarez-Montoya, J.; Marques, C. Leal-Junior, A.G.; Li, Y.; Carva-jal-Castrillón, A.; Lu, P.; Liu, C.; Gryllias, K.; and Zhang, Y. Among the most cited research is the work of Sierra-Perez, J.; Alvarez-Montoya, J.; and Carva-jal-Castrillón, A., where a health and usage monitoring system (HUMS) was developed and implemented in an unmanned aerial vehicle (UAV), based on 20 fiber Bragg gratings (FBG) embedded in the composite forward spar of the aircraft wing, a miniaturized data acquisition subsystem to collect stress signals, and a wireless transmission system for remote sensing, where flight data were used to validate a detection methodology using machine learning algorithms [41].

On the other hand, there are the least cited authors, but the most productive ones, since they are the ones who have made the most publications on the subject of study. Among these authors are Huagn, J.; Gerald, R.E.; and Zhu, C. For this group of authors, one of the most cited studies is that of Huagn, J., Gerald, R.E., and others, which is about the development of a smart helmet with a single integrated FBG sensor for real-time detection of blunt force impact events in helmets, using transient signals that provide information about both the magnitude and direction of the impact event, and using the data to train machine learning (ML) models [42].

As for the journals, Figure 4 shows the journals that have published on the topic of fiber optic sensors and machine learning, the figure shows that, the group in yellow corresponds to the journals with more publications and more citations, the group in blue corresponds to the journals with more citations, but no more publications, and the group in green corresponds to the most cited journals, but with fewer publications. Below, the technical aspects and contributions of some of the most relevant papers are highlighted according to the journal in which they were published. Journals with the most cited articles include: IEEE Sensors Journal, Sensors, and Journal of Lightwave Technology. One of the most cited articles is found in IEEE Sensors Journal and is about the research titled “Optical Fiber Specklegram Sensors for Mechanical Measurements: A Review”, whose authors are also among the most cited [1]. Another of the most cited investigations, found in the Journal of Lightwave Technology, is the paper entitled “Machine Learning for Turning Optical Fiber Specklegram Sensor into a Spa-tially-Resolved Sensing System. Proof of Concept,” which was published in 2018 and is one of the most representative articles on the topic [38].

Another article that has been widely cited in other studies was published in the IEEE Sensors Journal and is about the research titled “Blood Pressure Estimation Using Photoplethysmogram Signal and Its Morphological Features”, where the researchers present a machine learning model to estimate a person’s blood pressure (BP) using their photoplethysmogram (PPG) signal [43]. Next, there are the journals that, although they do not constantly publish research related to fiber optic sensors and machine learning, have some of the research that has had the greatest reference since they are among the most cited. Among these journals, they find each other: Mechanical Systems and Signal Processing, Journal of Neural Engineering and optical Fiber Technology. Among the most cited works, the research entitled “In-flight and wire-less damage detection in a UAV composite wing using fiber optic sensors and strain field pattern recognition”, published in the journal Mechanical Systems and Signal Processing, has already been mentioned since its authors are among the most cited. Next is the paper “A semi-supervised Support Vector Data Description-based fault detection method for rolling element bearings based on cyclic spectral analysis”, also published in Mechanical Systems and Signal Processing. This paper presents a canine-specific deep learning convolutional neural network (CNN) system for seizure prediction from ambulatory intracranial electroencephalogram (EEG) (iEEG) using a Mayo Epilepsy Assistant [44].

On the other hand, there are journals that have the largest number of publications on the topic but are not widely cited. These journals include Optics Express and the International Measurement Confederation’s Measurement Journal. Among the most relevant publications, the article entitled “Micro-scale fiber-optic force sensor fabricated using direct laser writing and calibrated using machine learning”, published in the journal Optics Express, presents a sensor of microscale fiber-optic force fabricated using direct laser writing (DLW) and for which calibration and optimization are performed, with a particular focus on data analysis using linear regression or artificial neural networks [6].

In terms of productivity by country, Figure 5 lists the countries that have made academic contributions on the topic of fiber optic sensors and machine learning, the figure shows that, the group in yellow corresponds to the countries with more publications and more citations, the group in blue corresponds to the countries with more citations, but no more publications, and the group in green corresponds to the most cited countries, but with fewer publications. Among the countries that can be considered references on the subject, the United States and China are the ones that concentrate the largest number of publications and those that have been most cited. One of the most representative studies is published in the journal Sensors and has 53 citations. This work was developed between researchers from the United States and China. In this research, the authors propose a leak-triggered wireless sensor network method based on machine learning to reduce the power consumption of the wireless sensor network and effectively prolong the system life cycle [45].

Then there are countries that have not produced highly cited publications in the field of study but have several publications in the field of knowledge. These countries include Brazil, Italy, India, Poland, Germany, Russia, and Japan. Among the most relevant publications is the work of [46] carried out by researchers from Italian institutions, which deals with a design and calibration methodology based on numerical modeling of the finite element method (FEM) for the development of a sensor soft touch capable of simultaneously resolving the magnitude and location of application of a normal load applied to its surface. The sensor design involves the integration of a fiber optic Bragg grating sensor in a Dragon Skin 10 polymer brick. Another study that is among the most relevant is that carried out by researchers from institutions in Brazil and Portugal; it is the work entitled “Optical Fiber Specklegram Sensors for Mechanical Measurements: A Review” [1].

## 4. Discussion

### 4.1. Analysis of Thematic Evolution

Theragnostics was the most-used term in 2015. In this example, sparse ML approaches were used with fiber optic sensors to detect cancer populations [47]. Two years later, the most common search term was wireless sensor, where researchers showed how to use step and walk analysis to identify a person’s walking gait from channel state information and how to use this information to identify a person using artificial intelligence (AI) methods [48], as well as for the detection of different substances and elements in water [49], later they began research approximately in 2014 where the use of sensors was combined using support vector regression by least squares [50].

In both 2018 and 2019, pattern recognition was by far the most common search term. This technology was used by several authors in conjunction with fiber optic sensors to improve the performance of these sensors. For example, a damage detection system based on strain field pattern recognition using FBGs, nonlinear principal component analysis, and clustering algorithms was developed and experimentally evaluated [51]. In 2020, the most frequently used word was structural health monitoring (SHM). In fact, several techniques have been investigated to monitor the structural health of bridges [52], steel pipelines [53], rolling element bearings [54], UAVs [41], and others. FBGs have been the most researched optical fiber sensing method this year, sometimes embedded in composite materials to increase resistance and protection during implementation. On the other hand, various ML techniques, such as CNN [52,55], self-organizing maps [41], semi-supervised support vector data description (SVDD) [54], and support vector machines (SVM) [52].

Fiber optic sensor became the most-used term in 2021. There was a significant amount of interest in the application of this type of sensor to measure a variety of different physical and chemical parameters, such as fluid flow estimation [56], mechanical impact [57], refractive index [42,58], liquid identification [59], gas detection [60], and so on. FBS was also widely used in this year [57,61]. However, other techniques such as Fabry–Perot interferometer [60], Brillouin optical time domain analyzer (BOTDA) [62], phase-sensitive optical time domain reflectometry (φ-OTDR) [51], and photonic crystal fibers (PCFs) [63] were also implemented in combination with ML techniques to optimize the measurements.

Finally, the most frequently used phrases in 2022 and 2023 were gold nanoparticles and SHM, respectively. The first used ML approaches to discover and categorize local chemical environments in self-assembled monolayer-protected gold nanoparticles [64]. These models were also used to predict the physicochemical properties and behavior of these nanoparticles in biological contexts [65]. On the other hand, several models using artificial neutral network (ANN), long short-term memory (LSTM), and pattern recognition algorithms have been developed to improve the response of fiber optic sensors when this technology is used to monitor the structural health of buildings [66,67]. In addition, ML techniques have been proposed to identify, localize, quantify, and visualize cracks through intelligent interpretation of strain distributions measured by distributed fiber optic sensors [3]. Similarly, ML techniques have been used to detect fatigue [68], vibration [69], and stress [9] in mechanical and civil structures.

An analysis of the evolution of topics in the scientific literature on the application of fiber optic sensors and machine learning reveals a constantly evolving and diversifying panorama. Initially, the focus was on detecting water leaks in underground infrastructure, but over time, it has shifted to more specialized and diverse areas, such as structural health monitoring and nanoparticle characterization. This diversification indicates a maturation of the field, with greater sophistication in the integration of machine learning techniques into the design and operation of optical sensing systems. This technological evolution reflects not only progress in research but also adaptation to the needs of different sectors, such as medicine, civil engineering, and industry.

In the scientific literature, this shift in focus implies an increase in knowledge and interdisciplinary collaboration among researchers from different fields. Advances in fiber optic sensors and machine learning provide new opportunities to solve complex problems and address challenges in various fields of study.

These trends indicate potential areas for future research. The use of machine learning in nanoparticle characterization shows potential in biomedicine and nanotechnology. In addition, advanced machine learning techniques can help monitor and diagnose structural problems in civil and mechanical infrastructure, leading to new research opportunities at the intersection of optics, computing, and engineering. These areas of future research offer opportunities for innovation and scientific collaboration in the coming years.

In addition, Figure 6 does not include the years 2013 and 2016 because no dominant term or concept was identified during this time period. The circular graph was chosen to provide a more intuitive visualization of the evolution of the field, allowing for a clear representation of the distribution of key terms over time and facilitating the identification of trends and patterns in the scientific literature on the topic.

### 4.2. Analysis of Keyword Co-Occurrence

The most common phrase in most studies of fiber optic sensors integrated with AI, as shown in the keyword network (Figure 7) where the different colors indicate the grouping by group of words, is leak detection and its association with DL. For example, in the research by Shiloh et al. [70], the authors generate a training set for DAS using generative adversarial net methodology. In another study, the authors propose a method based on ensemble learning of CNN for predicting the fault location of a Sagnac distributed fiber optic sensor system. The proposed method achieves accurate prediction of an arbitrary fault position with a mean absolute error of no more than 14.6 m and a location resolution of 10 m [71]. In addition, an SVM-based pipeline leak detection and prewarning system is presented by Qu et. al [72]. The authors use an optical fiber sensor to monitor a pipeline, and the SVM analyzes vibration signals based on features to identify whether an irregular event is occurring. When an unusual event is observed along a pipeline, the exact location is determined using this method.

DL is one of the most widely used terms in this field of research, as various studies report the use of data to optimize fault location [72], detect earthquakes [41], predict pipeline degradation [73], identify bending direction [67], detect damage [50], and estimate crowd flow on pedestrian bridges [51].The authors of each research used deep learning (DL) approaches such as CNN, recurrent neural networks (RNNs), or deep autoencoders to evaluate data collected by fiber optic sensors and extract useful information for the specific application. The widespread use of DL methods in these different applications demonstrates their adaptability and effectiveness in evaluating data collected by fiber optic sensors. DL algorithms excel at recognizing complicated patterns and correlations in large data sets, making them a great tool for extracting significant insights from the large amount of data received by fiber optic sensors [74,75,76].

In this search for fiber optic sensors, research has focused on providing solutions using artificial intelligence and machine learning in various fields such as healthcare and structural health monitoring, since the integration of these technologies helps to develop more efficient and intelligent systems for monitoring and analysis. In the field of SHM, the integration of fiber optic sensors with artificial intelligence and machine learning techniques has revolutionized the way we assess and monitor the health and integrity of structures such as bridges, buildings, and pipelines [77,78,79,80]. Fiber optic sensors can provide distributed and real-time measurements of strain, temperature, and vibration, enabling continuous monitoring of structural behavior. By incorporating artificial intelligence and machine learning algorithms, the vast amount of data collected by these sensors can be processed and analyzed to detect structural anomalies, predict potential failures, and optimize maintenance strategies.

Finally, the integration of artificial intelligence and machine learning with fiber optic sensors in healthcare and structural health monitoring represents a paradigm shift in how we use sensor data to improve outcomes. These technologies enable real-time monitoring, early detection of anomalies, predictive maintenance, and data-driven decision making. As research and technology advances, the synergy between fiber optic sensors and AI/ML will continue to drive innovation in these areas, leading to more efficient, cost-effective, and sustainable solutions for healthcare and infrastructure monitoring.

### 4.3. Analysis of Emerging Words

The validity quadrants for various topics in optical fiber sensing and machine learning research are shown in Figure 8. The most commonly used term in quadrant I is optical fiber sensor, which refers to the use of optical fiber technology to implement various solutions for measuring chemical and physical properties such as temperature [64,81,82,83,84], strain [44,53,85,86,87], refractive index [88,89,90,91,92,93,94], curvature [70,71], vibration [72], and so on. Similarly, additional terms such as structural health monitoring, deep learning, FBGs, and leak detection have been identified as the most frequently used terms in the last three years. These terms are associated with research on predicting cracks, degradation, and mechanical and structural failures of buildings [46,47,71], pipelines [74], UAVs [72], and others. In these cases, the authors used machine learning and deep learning models to improve the accuracy and efficiency of data analysis and prediction. For example, by applying ML techniques, researchers were able to extract valuable insights from the vast amount of data collected by fiber optic sensors, enabling them to detect and predict potential problems in real time [41,46,47]. In addition, the integration of FBGs with DL algorithms has further improved the accuracy and reliability of these predictive models.

Leak detection has emerged as another critical area of research in fiber optic sensing and ML. By deploying fiber optic sensors in pipelines and other fluid transport systems, researchers have been able to detect and locate leaks in a timely manner. Machine learning techniques have played a critical role in analyzing the sensor data, identifying patterns indicative of leaks, and distinguishing them from normal operational variations. This has greatly improved the efficiency of leak detection and minimized the potential for costly damage and environmental hazards [53,64].

Quadrant II includes the most recent and common keywords, such as gold nanoparticles [92,93,94]. In this case, the ML model has been used to quantify gold nanoparticles using supervised ML techniques [92], to analyze transmission electron microscope images of metal nanoparticles [91], and to use gold nanoparticles as biosensors [94,95]. This indicates a growing interest in the application of optical fiber sensing combined with ML for nanotechnology-related research. The terms distributed optical fiber sensor, DAS, artificial intelligence, and extreme learning machine are found in this quadrant. The first two terms are related because they both involve the concept of using fiber optic technology for distributed sensing applications. The first refers to systems where the fiber itself serves as the sensing element along its entire length. This allows the development of a continuous and spatially resolved sensor network [44,46,92]. The other is a subset of distributed fiber sensing that focuses on the detection and analysis of acoustic waves or vibrations. The optical fiber in DAS systems acts as a distributed microphone, converting acoustic signals into optical signals that can be studied and understood [52]. The combination of distributed fiber optic sensors with machine learning techniques such as the extreme learning machine has proven to be very effective. ML algorithms can be trained to evaluate the massive amounts of data generated by distributed sensors, identify anomalies, and extract useful patterns. Real-time monitoring and early warning systems are now possible in applications such as structural health monitoring, security surveillance, and industrial process management.

Left ventricular assist devices (LVAD) and mild traumatic brain injury are terms in Quadrant II. These terms refer to key areas of research where fiber optic sensing and machine learning have been used to solve critical healthcare problems [31,96,97]. For example, fiber-sensing technology has been used to improve LVAD monitoring and control. Researchers can collect real-time data on many factors such as blood flow, pressure, and temperature by incorporating optical fibers into the device. A CNN algorithm was then used to analyze the data, allowing early detection of likely problems or anomalies, optimizing device performance, and improving patient outcomes [97]. Thus, the combination of fiber optic sensing and machine learning in LVAD and mild traumatic brain injury research holds great promise for advancing personalized medicine and improving patient care. Clinicians can gain deeper insights into patient-specific characteristics by using machine learning models trained on large datasets, allowing for more accurate diagnosis, prognosis, and treatment planning. In addition, fiber optic sensors and machine learning could provide useful longitudinal data for research and the development of new treatments and interventions.

On the other hand, the terms leak location, polymer optical fiber, fiber optic sensor, and concrete were also found in this quadrant. In this case, all these keywords are related because, as mentioned before, optical fiber sensors have been widely explored to monitor the structural health of buildings and pipelines due to the many advantages that this type of sensor offers compared to traditional technologies such as wireless sensors, MEMS, and others [9,26,38,43,64]. Similarly, POFs have emerged as a promising alternative to traditional silica-based optical fibers. POFs offer several advantages, including flexibility, ease of installation, and compatibility with curved surfaces. These characteristics make them well suited for applications such as structural health monitoring, where they can be embedded in concrete structures to monitor strain, temperature, and other parameters. The combination of POFs and fiber optic sensors enables efficient and cost-effective monitoring of the structural integrity of buildings and other infrastructure. In fact, this type of optical fiber has been used in combination with ML models for gesture recognition [98], determining the elastic properties of printed fiber-reinforced polymers [99], and assessing delamination in fiber-reinforced polymer composite beams [100]. The term neural network is found between quadrants II and III. NN has been implemented in various assistive techniques for damage detection [24,28], pipeline degradation monitoring [51], or bending direction determination [42].

Pattern recognition, monitoring, wireless sensor network (WSN), and fault detection are examples of less common and contemporary words found in Quadrant III. The process of identifying and categorizing patterns within a given data set or system is referred to as pattern recognition. Pattern recognition is critical in the context of fiber sensing and machine learning research for analyzing sensor data and extracting useful information [24,28,42].

Monitoring, fault detection, and WSNs are other terms closely related to fiber optic sensing and machine learning. They refer to the continuous observation and measurement of various parameters or conditions. Fiber optic sensors, with their ability to provide real-time and distributed sensing capabilities, enable efficient monitoring of a wide range of applications, including structural health monitoring, environmental monitoring, and industrial process monitoring. The inclusion of pattern recognition, monitoring, wireless sensor networks, and fault detection in Quadrant III reflects ongoing research efforts to develop advanced techniques for analyzing and interpreting data collected by fiber optic sensors. Machine learning, with its ability to handle large data sets and identify complex patterns, is a key enabler of this research. By using ML algorithms, researchers can unlock the full potential of fiber optic sensor systems, enabling more efficient and intelligent monitoring, fault detection, and decision making. Finally, there are no components in Quadrant IV, which consists of the least and most recently used terms.

### 4.4. Classification of Keywords

Taking into account the keywords, it is possible to catalog the keywords in four dimensions corresponding to the most used techniques, the main tools, the applications, and the main characteristics, as can be seen in Table 1. Next, the summary of the variables is presented, such as the main trends according to the indicated categories, considering the results obtained for the agenda, network, cluster, and classification according to the year of the keywords.

### 4.5. Research Agenda

Figure 9 shows a complete research program for fiber sensing and ML integration. The agenda is organized chronologically by year and is based on the relevance and importance of different research concepts. It provides an overview of the changing trends and topical interests in this multidisciplinary field. The agenda is a useful resource for scientists to guide them in selecting critical areas of focus and exploring new avenues for progress in fiber optic sensing and machine learning.

Research on topics such as monitoring, fault detection, WSN, BOTDA, Brillouin scattering, NN, and damage detection as central topics of discussion leaves the research agenda and falls behind in the discussion on fiber optic sensing, although the discussion on NNs [63,101], artificial intelligence [102], and damage detection are underexplored [28,103]. On the other hand, between 2014 and 2019, various terms such as fiber optic sensor, FBGs, monitoring, and fault detection were frequently used. Only the first two are still receiving attention, while the latter two have fallen off the academic agenda for the reasons described above. These concepts have been widely used in research with a wide range of objectives. One prominent application was the use of clustering algorithms to develop a damage detection tool based on strain field pattern recognition and FBGs [24]. By harnessing the power of clustering algorithms, researchers were able to categorize and identify different strain patterns acquired by the FBGs, enabling real-time identification of structural damage. Machine learning techniques have also been used to calibrate sensors to improve their optical performance [104], to detect respiratory rate using a wavelet classifier that analyzes the signal collected by an FBG-based vital signs sensor [54], and to localize impacts in composite structures [41].

Other terms such as neural networks, damage detection, Brillouin scattering, BOTDA, leak detection, artificial intelligence, DL, and pattern recognition became relevant from 2019 to 2022; they are still widely discussed but no longer valid. Research has been done on these topics, including temperature measurement [24], continuous detection of heart rate in signals using DL [36], leak detection in water pipes based on random forest [64], identification of liquids using a simple fiber optic tip sensor, and a pre-trained CNN [34]. In addition, some of the more recent work focused on telecommunications and signal processing in the transmission medium, specifically cognitive radio, has made significant contributions to the research. They have addressed issues ranging from the confrontation of support vector machines against probabilistic Byzantine attacks to cooperative spectrum sensing in cognitive radio networks (CRN). In this context, the authors use support vector machines (SVMs) to identify malicious secondary users (MSUs) by providing a maximum margin hyperplane. In particular, the characteristics of the data generated for spectrum detection benefit from the state of the primary unit (PU) in the training process [18].

Another recent study focusing on spectrum sensing and data security in this domain has focused on addressing the limitations of labeled data in cognitive radio (CR) networks. A two-stage semi-supervised learning framework is proposed for cooperative spectrum sensing (CSS) against spectrum sensing data falsification (SSDF) attacks. This approach combines the outstanding effectiveness of semi-supervised support vector machines (S3VM) with the fast convergence of K-means [105]. Furthermore, a work presented by the authors [106] aims to address the challenges faced by secondary users (SU) against spectrum sensing data falsification (SSDF) attacks launched by malicious users (MU). In this context, they propose an algorithm designed to calculate the degree of suspicion (DE) for each SU, using the concept of average suspicion degree (ASD), and to adjust the weights of SUs according to their ASD [107]. In another study focusing on spectrum data security, a scheme based on a support vector machine (SVM) is introduced to analyze the behavior of secondary users based on multiple rounds of energy values. This provides an accurate classification evaluation index, which allows better adaptability in detecting malicious secondary users (MSU) in different scenarios [108].

Finally, the research agenda focuses on topics such as distributed fiber optic sensors, as demonstrated by the use of this technology to produce a new generation of fiber optic sensors in various contexts [9,44,53]. For example, a novel pipe-in-pipe (PIP) leak detection system using distributed temperature sensing (DTS) with ML has been proposed and experimentally validated [43]. Similarly, machine learning and intelligent interpretation of recorded strain distributions from dispersed fiber optic sensors have been used to automate crack detection, localization, quantification, and visualization [44]. Polymer optical fibers (POFs), gold nanoparticles, and CNN are also on the research agenda. The studies have been used in research to identify the antibiotic resistance phenotype [57], estimate the degradation of composite structures [45], and monitor directional bending [42], among others.

The proposed research program in Figure 9 provides a comprehensive view of the key research areas in the integration of fiber optic sensing and machine learning. However, opportunities for future development are identified, particularly in the area of machine learning applications in fiber optic sensing. Although topics such as fault detection, WSN, and techniques such as BOTDA and Brillouin scattering have declined in importance on the research agenda, there is clearly continued interest in the application of machine learning algorithms to fault detection. structural damage, and defects.

In the period between 2019 and 2022, terms such as neural networks, damage detection, Brillouin sparsity, BOTDA, leak detection, artificial intelligence, deep learning, and pattern recognition have emerged as relevant and widely discussed topics. Although some of these terms are no longer at the forefront of research, there is still considerable interest in their application in the context of fiber optic sensing. Research has been conducted to improve temperature measurement, continuously detect heart rate using deep learning, identify leaks in water pipes using random forests, and discriminate liquids using fiber optic tip sensors and pre-convolutional neural networks.

In addition, the future research program focuses on topics such as distributed fiber optic sensors, where new applications have been proposed using this technology in various contexts. For example, a pipeline leak detection system using distributed temperature sensors with machine learning has been proposed and experimentally validated. Similarly, machine learning and intelligent interpretation of strain distributions recorded by distributed fiber optic sensors have been used to automate crack detection, localization, quantification, and visualization. In addition, future research will focus on the use of polymer optical fibers, gold nanoparticles, and convolutional neural networks for applications such as antibiotic resistance phenotype identification, composite degradation estimation, and flexure monitoring, among others.

### 4.6. Limitations

One of the limitations identified in this research is the restriction imposed by the search criteria used in the publication databases. The search was limited by the selection of specific terms, such as “machine learning” and “optical sensors”, which may have excluded relevant papers using different terminology. The variety of terms and abbreviations used in the field of machine learning and optical sensing may have hindered the comprehensiveness of the search, as practitioners may use a wider range of terms and acronyms in their publications, such as “neural network”, “artificial intelligence”, “CNN”, and “DAS”, among others.

Another important limitation that needs to be addressed is the lack of consideration of the distinction between scientific papers that focus on the development of fiber optic sensors and those that focus on the application and use of these sensors in specific contexts. This distinction is critical to understanding the contribution of each paper to the literature and to identifying areas of research that require further attention. Failure to make this distinction can lead to a lack of clarity in the literature review and biased interpretation of results, limiting the ability of future researchers to build on existing knowledge and make meaningful contributions to the field. To overcome this limitation, future research must carefully consider the categorization of scientific papers to ensure that both the development of new sensors and their application in different contexts of use are recognized. This will allow for a more complete and accurate understanding of the scientific literature and facilitate the identification of emerging areas of research and opportunities for advancement in the field of optical sensors.

Among the major limitations, there are still significant gaps regarding the use of materials in the design of optical fibers. It is crucial to note that no study trends have been observed in the use of innovative materials, such as graphene, for the configuration of fiber optic sensors. This lack of research in the area of new materials suggests an area of opportunity that could be explored in future scientific studies and research in the field of optical fiber technology. Considering and experimenting with materials beyond current conventions could reveal unique and advanced properties, thereby promoting the continued development and improvement of fiber optic-based optical sensor technology.

## 5. Conclusions

The joint progress in the use and characterization of materials has allowed progress in the use of devices for different applications in contexts that were not thought of a few years ago; microelectronics, metamaterials, flexibility, and miniaturization have led to the focus of different technologies on applications and areas of different contexts beyond just electronics, mechanics, or telecommunications. In recent years, advances in software and the integration of technologies such as artificial intelligence in different fields have made it possible to digitize and drive the development of new applications or improve the performance of existing ones. Since 2018, the growing interest in integrating different machine learning techniques in fiber optic sensor applications has allowed us to obtain better results in their performance.

Consistency was found among the main references on the topic, with the most cited authors belonging to institutions located in the countries that also present the most cited publications. Similarly, the most cited journals contain the publications of authors with the highest quality indicators. In this way, research in fiber optic sensors has moved from a focus on multiplexing techniques to improve wireless sensor networks to a focus on pattern recognition to improve performance in structural health monitoring.

Regarding the clusters of key terms, the main network is headed by structural health monitoring, preceded by terms such as pattern recognition, convolutional neural networks, and artificial neural networks, which are the main techniques that have been used in research on the use of fiber optic sensors for health monitoring. The next cluster is led by deep learning and is preceded by terms such as fiber Bragg gratings and fiber optic sensors. This network mainly refers to the use of wireless sensors in infrastructure.

The next cluster is led by the term leak detection, which refers to the use of sensors in gas leaks. Finally, in terms of analyzing the validity of the key terms and the research agenda, they guide future research towards proposing new ways of using fiber optic sensors in leak detection and structural health monitoring. The previous applications, mainly from the use of deep learning techniques and the application of fiber Bragg gratings.

In conclusion, the integration of machine learning techniques into fiber optic sensors has great potential to improve the accuracy and efficiency of these devices. However, some areas still require further research and development. Additional research is needed to understand how different machine learning techniques can optimize the detection and monitoring of physical and chemical parameters using optical sensors. To advance the field, it is critical to explore new applications and use scenarios for these sensors, as well as to investigate methods to overcome the technical and operational challenges that still exist in their large-scale fabrication and deployment.

This requires interdisciplinary collaboration between experts in optics, machine learning, materials engineering, and other relevant fields. It is essential to identify and address current bottlenecks in the integration of these technologies. It is also important to establish standards and protocols for evaluating and validating new approaches and devices. In addition, further studies are needed to explore how artificial intelligence can optimize not only the accuracy and sensitivity of sensors but also their robustness, interoperability, and adaptability to changing environments and adverse conditions. By addressing these issues, we can fully realize the potential of machine-learning-based fiber optic sensors. This will enable us to expand their applications in diverse fields, including environmental monitoring, medicine, and security.

## Figures and Tables

**Figure 1 sensors-24-02200-f001:**
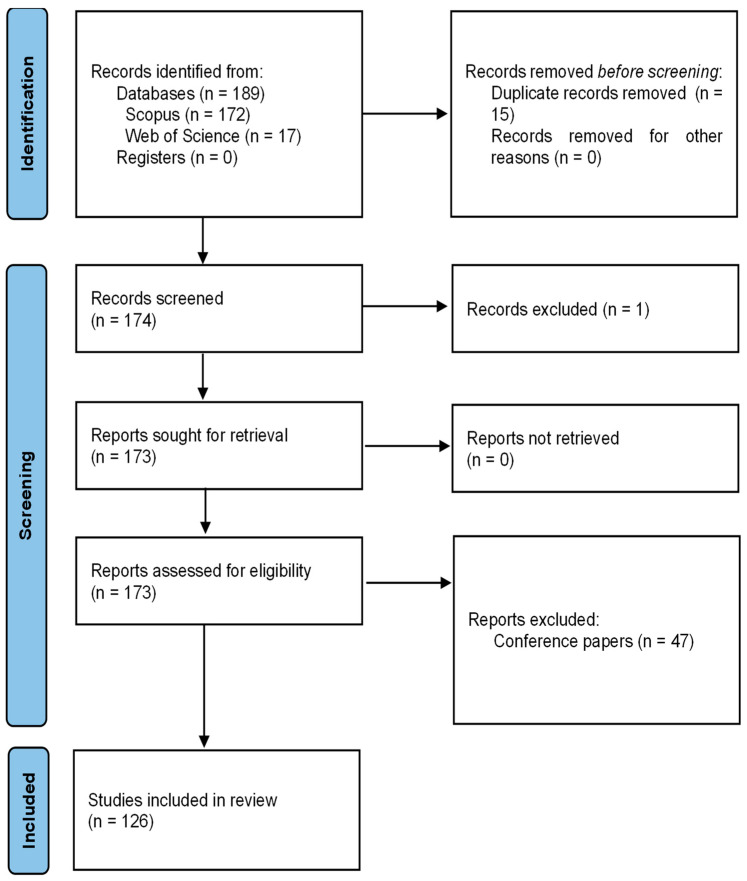
PRISMA flow chart. Own elaboration based on Scopus and Web of Science.

**Figure 2 sensors-24-02200-f002:**
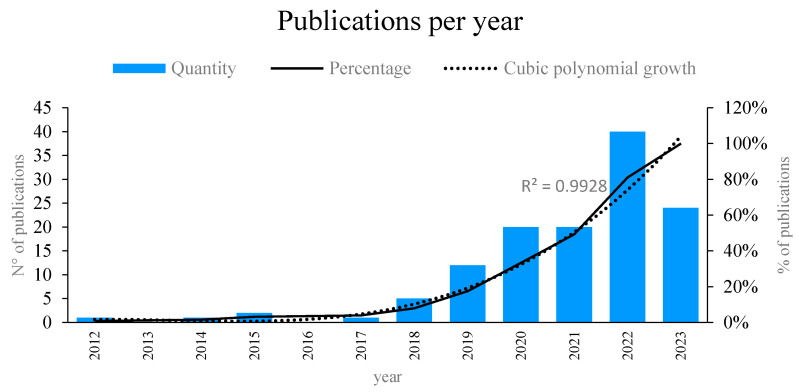
Quantity and percentage of publications per year. Own elaboration based on Scopus and Web of Science.

**Figure 3 sensors-24-02200-f003:**
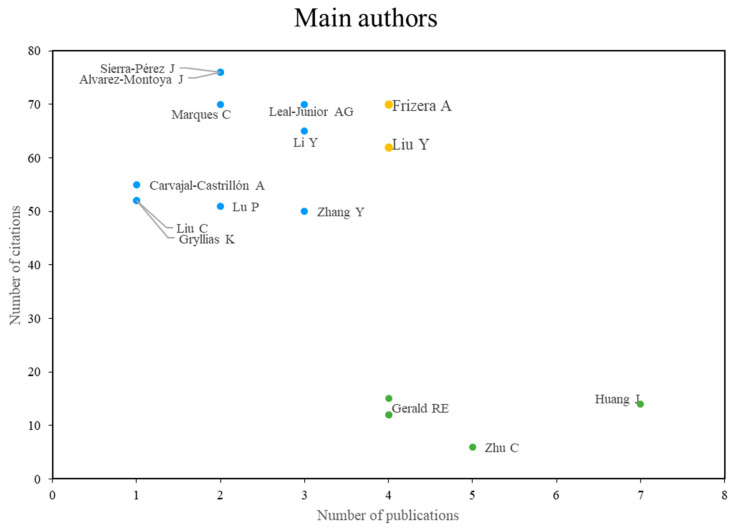
Main authors according to the number of citations vs. the number of publications. Own elaboration based on Scopus and Web of Science.

**Figure 4 sensors-24-02200-f004:**
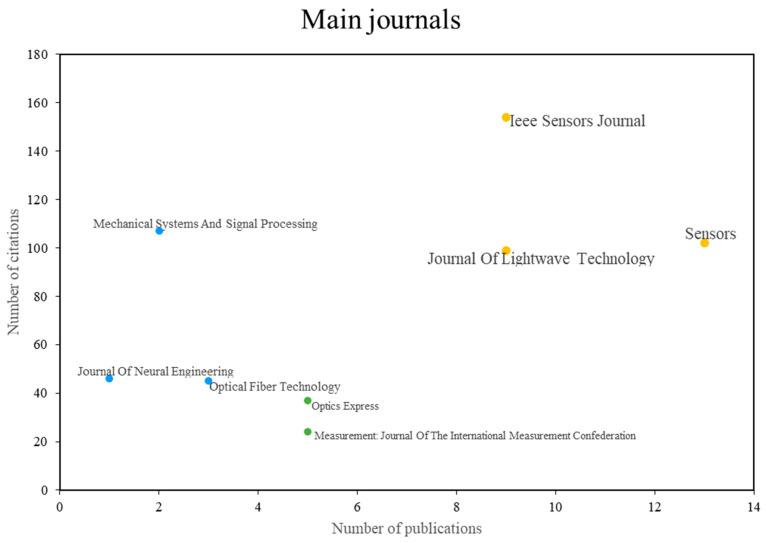
Major journals according to the number of citations vs. the number of publications. Author’s elaboration based on Scopus and Web of Science.

**Figure 5 sensors-24-02200-f005:**
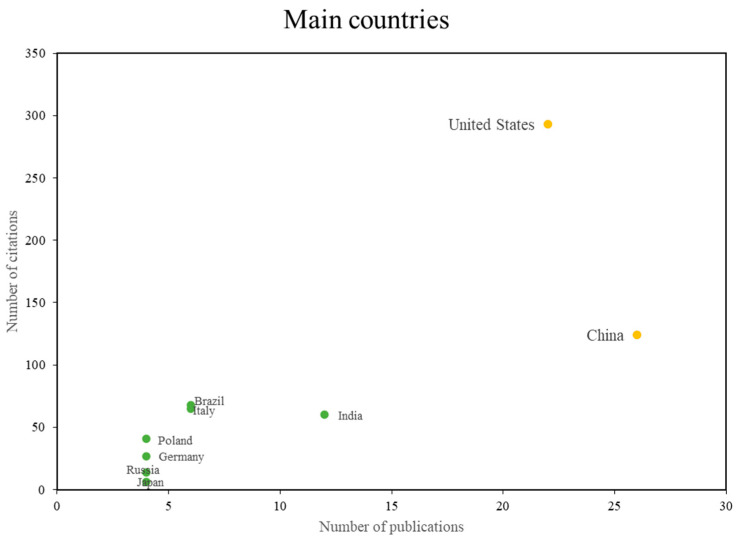
Main countries according to the number of citations vs. the number of publications. Author’s calculations based on Scopus and Web of Science.

**Figure 6 sensors-24-02200-f006:**
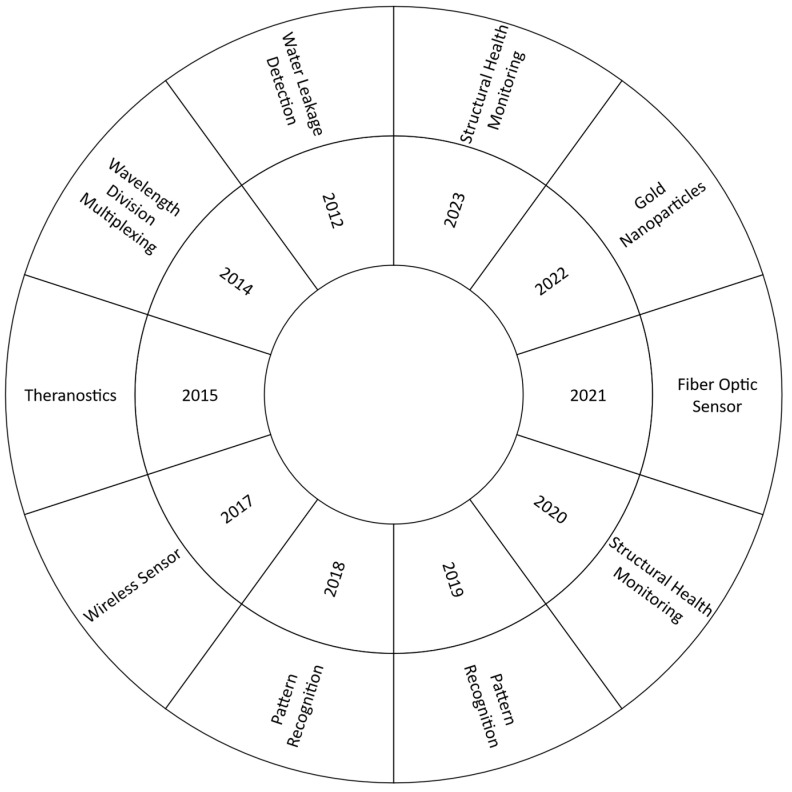
Keywords by year for optical fiber sensors and machine learning techniques. Source: the authors.

**Figure 7 sensors-24-02200-f007:**
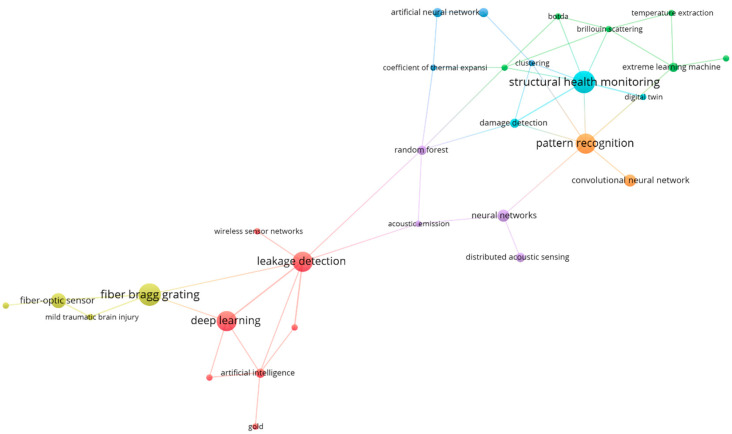
Keyword co-occurrence network according to their main relationship node. Own elaboration based on Scopus and Web of Science.

**Figure 8 sensors-24-02200-f008:**
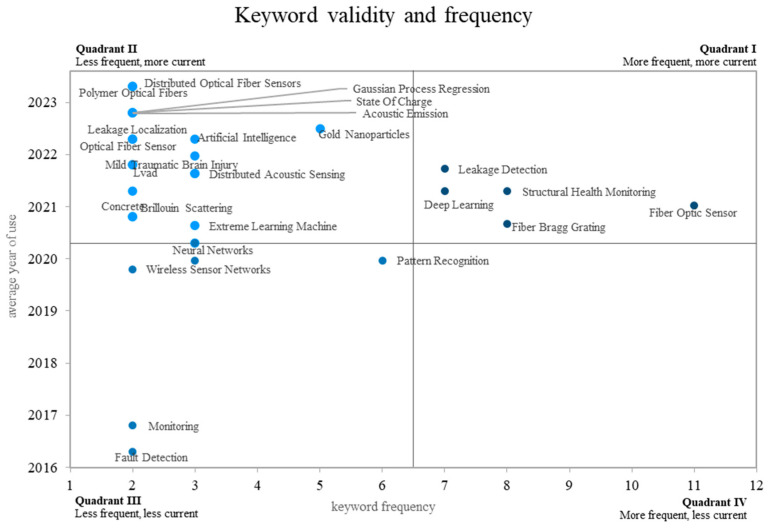
Validity and frequency of keywords per year versus the frequency of appearance of each word. Own elaboration based on Scopus and Web of Science.

**Figure 9 sensors-24-02200-f009:**
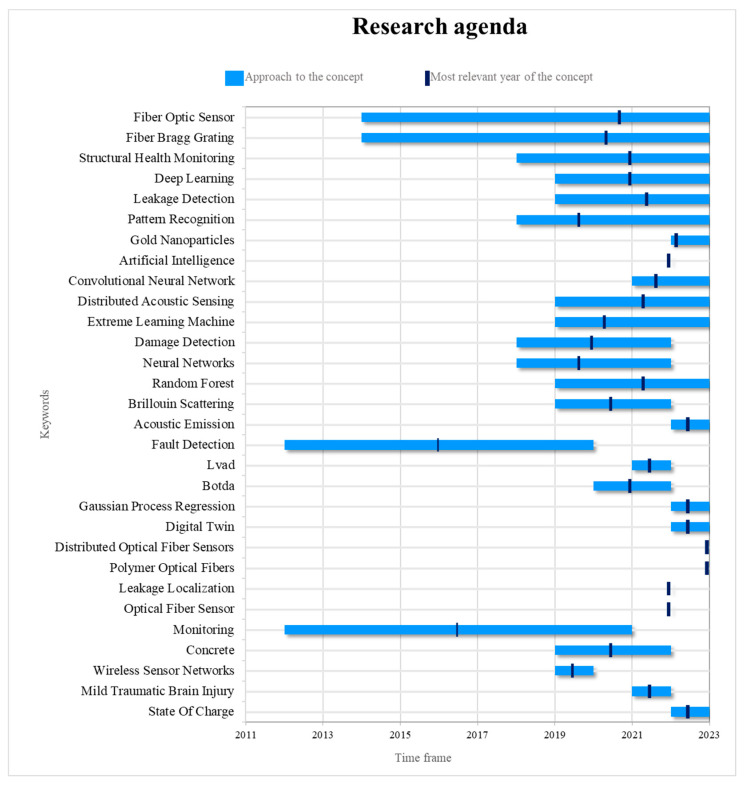
Research agenda for fiber optic sensing and machine learning based on the utilization of the concept during the mentioned period, highlighting the year of highest frequency. Source: the authors.

**Table 1 sensors-24-02200-t001:** Keyword classification according to function.

Techniques	Tools	Applications	Characteristics
Deep Learning	Fiber Bragg Gratings	Leak detection	Temperature
Deep Learning Pattern Recognition	Distributed Fiber Optic Sensors	Structural Health Monitoring	Curvature
Convolutional Neural Networks	Data Analysis and Prediction	Left ventricular assist devices	Refractive Indicator
Artificial Neural Networks	Efecto Vernier	Crack prediction	Strain
Random forest	Fiber Specklegram Sensors	Mechanical and structural building failures	Acoustics/Vibrations
		Predicting problems	Deformation

Source: own elaboration from the keywords.

## Data Availability

The data may be provided free of charge to interested readers by requesting the correspondence author’s email.

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
