# Peer review of "Machine Learning Applications in Optical Fiber Sensing: A Research Agenda"

_sensors, 2024, doi:10.3390/s24072200_

Round 1
Reviewer 1 Report
Comments and Suggestions for Authors
(1) Full name of the abbreviations should be presented, such as DTS、3-D FRP-FBG、FBS
(2) Why the abbreviation of Long-Term Memory is LSTM?
(3) Typing errors should be avoided, such as ooptical Fiber Technology
(4) What are the most frequently used phrases in 2013 and 2016?
Comments on the Quality of English LanguageModerate editing of English language required
Author Response
February 25th, 2024
Dear
Sensors
Kind regards
In accordance with the suggestions of the reviewers in our article “Machine Learning Applications in optical fiber sensing: A Research agenda”, the following changes were made, properly marked with red letters in the article:
|
Reviewer |
Comment |
Answer |
|
Reviewer 1 |
(1) Full name of the abbreviations should be presented, such as DTS、3-D FRP-FBG、FBS |
A section is added with all the acronyms used in the article, at the beginning of the writing |
|
Reviewer 1 |
(2) Why the abbreviation of Long-Term Memory is LSTM? |
The correction is made, since the correct name is Long Short Term Memory (LSTM) |
|
Reviewer 1 |
(3) Typing errors should be avoided, such as ooptical Fiber Technology |
The aforementioned correction is made |
|
Reviewer 1 |
(4) What are the most frequently used phrases in 2013 and 2016? |
A paragraph is added after Figure 6, explaining why the years 2013 and 2016 are not available. |
|
Reviewer 1 |
Moderate editing of English language required |
A syntax and spelling check is carried out on the entire article. |
We look forward to your comments and hope to hear from you soon.
Thank you very much
_
The authors
Reviewer 2 Report
Comments and Suggestions for Authors
Comment 1
- The abstract lacks specific details about the methodologies employed in the bibliometric analysis, making it challenging to evaluate the rigor and reliability of the research trends identified.
While the abstract mentions the application of deep learning techniques and fiber Bragg gratings, it lacks clarity on the specific advancements or breakthroughs achieved in these areas.
It could benefit from providing a brief overview of the limitations or challenges identified in the existing literature, offering a more comprehensive perspective on the state of research in machine learning applications for optical fiber sensing.
Comment 2
- The introduction is excessively lengthy and contains repetitive information, leading to a lack of conciseness. A more streamlined presentation of key points would enhance the clarity and engagement of the reader.
The flow of the introduction is somewhat disorganized, making it challenging for the reader to follow the logical progression of ideas. Clearer structuring of information would improve the overall coherence of the text.
While providing context is crucial, the introduction seems to dwell extensively on general background information about sensors, potentially diluting the focus on the specific topic of machine learning applications in fiber optic sensors. It would benefit from a more direct and focused approach towards the research agenda.
Comment 3
- In section 2, the exclusion criteria, especially in the screening process, are briefly mentioned without providing a comprehensive explanation.
Comment 4
- The section on data management is concise and lacks elaboration on how the Microsoft Excel tool and VOSviewer will be specifically utilized.
Comment 5
- The discussion lacks critical analysis and interpretation of the findings presented in Figure 6. The section primarily describes the most frequently used terms over the years without delving into the implications, limitations, or potential areas for future research based on these trends.
Comment 6
- The conclusion could benefit from a more robust discussion on the future outlook of machine learning applications in fiber optic sensors. It briefly mentions guiding future research but falls short of providing concrete suggestions or areas that warrant further exploration.
Author Response
February 25th, 2024
Dear
Sensors
Kind regards
In accordance with the suggestions of the reviewers in our article “Machine Learning Applications in optical fiber sensing: A Research agenda”, the following changes were made, properly marked with red letters in the article:
|
Reviewer |
Comment |
Answer |
|
Reviewer 2 |
*- The abstract lacks specific details about the methodologies employed in the bibliometric analysis, making it challenging to evaluate the rigor and reliability of the research trends identified. While the abstract mentions the application of deep learning techniques and fiber Bragg gratings, it lacks clarity on the specific advancements or breakthroughs achieved in these areas. It could benefit from providing a brief overview of the limitations or challenges identified in the existing literature, offering a more comprehensive perspective on the state of research in machine learning applications for optical fiber sensing. |
The summary is adjusted by expanding the description of the methodology used and the limitations that were found in the field of study. |
|
Reviewer 2 |
* The introduction is excessively lengthy and contains repetitive information, leading to a lack of conciseness. A more streamlined presentation of key points would enhance the clarity and engagement of the reader. The flow of the introduction is somewhat disorganized, making it challenging for the reader to follow the logical progression of ideas. Clearer structuring of information would improve the overall coherence of the text. While providing context is crucial, the introduction seems to dwell extensively on general background information about sensors, potentially diluting the focus on the specific topic of machine learning applications in fiber optic sensors. It would benefit from a more direct and focused approach towards the research agenda. |
The introduction was readjusted, so that it was more concise and cohesive |
|
Reviewer 2 |
* In section 2, the exclusion criteria, especially in the screening process, are briefly mentioned without providing a comprehensive explanation. |
The explanation is expanded regarding the eligibility criteria |
|
|
|
and how the selection process occurred. |
|
Reviewer 2 |
The section on data management is concise and lacks elaboration on how the Microsoft Excel tool and VOSviewer will be specifically utilized. |
The explanation of data management is expanded, in relation to the elements suggested by the reviewer |
|
Reviewer 2 |
The discussion lacks critical analysis and interpretation of the findings presented in Figure 6. The section primarily describes the most frequently used terms over the years without delving into the implications, limitations, or potential areas for future research based on these trends. |
The interpretation of these results is expanded in light of their implications and potential areas for future research. |
|
Reviewer 2 |
The conclusion could benefit from a more robust discussion on the future outlook of machine learning applications in fiber optic sensors. It briefly mentions guiding future research but falls short of providing concrete suggestions or areas that warrant further exploration. |
Two new conclusions are added that allow us to mention what was recommended by the reviewer |
We look forward to your comments and hope to hear from you soon.
Thank you very much
_
The authors
Reviewer 3 Report
Comments and Suggestions for Authors
The need to constantly monitor and control various health, infrastructure and natural factors has led to the design and development of technological devices in a wide range of fields. This has led to the creation of different types of sensors that can be used in the monitoring and control of different environments, such as fire, water, temperature, movement, among others. These sensors detect anomalies in the input data to the system, allowing alerts to be generated, facilitating early attention to risks. However, the advent of artificial intelligence has led to the improvement of sensor systems and networks, resulting in devices with better performance and timely results by incorporating various features that increase their precision. Therefore, the objective of this paper is to identify research trends around the development of machine learning applications in fiber optic sensors through a bibliometric analysis using the PRISMA set. However, there are following comments:
1. The part of Introduction section is not sufficient. An introduction section generally includes the following: (1) Motivation for the work (2) Background/Technical information or terms (3) Problem being addressed (4) Existing major solutions and research gaps and (5) Proposal of this paper and its significance.
2. The authors are requested to provide all the tabular form of all the abbreviations provided in the paper.
3. Highlight the challenges or limitations of the existing approaches, the authors should provide a presentation in the introduction.
4. Provide more details about the future work about machine learning machine learning applications in optical fiber sensing.
5. The author should discuss or comment more on the research in related fields or methods in the past three years to highlight the innovation of their research work, i.e.,
[R1] Zhixuan Z, Jun W, Jipeng G, et al. Support Vector Machine Process Against Probabilistic Byzantine Attack for Cooperative Spectrum Sensing in CRNs[C]//Proceedings of the 2023 8th International Conference on Machine Learning Technologies. 2023: 269-276.
[R2] Chen Z, Wu J, Bao J. Semi-supervised Learning-enabled Two-stage Framework for Cooperative Spectrum Sensing Against SSDF Attack[C]//2022 IEEE Wireless Communications and Networking Conference (WCNC). IEEE, 2022: 554-559.
[R3] Cheng Z, Song T, Zhang J, et al. Self-organizing map-based scheme against probabilistic SSDF attack in cognitive radio networks[C]//2017 9th International conference on wireless communications and signal processing (WCSP). IEEE, 2017: 1-6.
[R4] Zhu H, Song T, Wu J, et al. Cooperative spectrum sensing algorithm based on support vector machine against SSDF attack[C]//2018 IEEE international conference on communications workshops (ICC workshops). IEEE, 2018: 1-6.
etc.
6. The discussion section should be presented in categories.
Comments on the Quality of English LanguageThe language of this paper is relatively fluent and there are no issues.
Author Response
February 25th, 2024
Dear
Sensors
Kind regards
In accordance with the suggestions of the reviewers in our article “Machine Learning Applications in optical fiber sensing: A Research agenda”, the following changes were made, properly marked with red letters in the article:
|
Reviewer |
Comment |
Answer |
|
|
Reviewer 3 |
1. The part of Introduction section is not sufficient. An introduction section generally includes the following: (1) Motivation for the work (2) Background/Technical information or terms (3) Problem being addressed (4) Existing major solutions and research gaps and (5) Proposal of this paper and its significance. |
The introduction is expanded by including new quotes and taking into account the different elements requested by the reviewer. |
|
|
Reviewer 3 |
2. The authors are requested to provide all the tabular form of all the abbreviations provided in the paper. |
All abbreviations of the article are tabulated at the beginning of the paper |
|
|
Reviewer 3 |
3. Highlight the challenges or limitations of the existing approaches, the authors should provide a presentation in the introduction. |
The explanation requested by the reviewer is added to the introductory section |
|
|
Reviewer 3 |
4. Provide more details about the future work about machine learning machine learning applications in optical fiber sensing. |
Three additional paragraphs giving further details are added at the end of the discussion section |
|
|
Reviewer 3 |
5. The author should discuss or comment more on the research in related fields or methods in the past three years to highlight the innovation of their research work, i.e., [R1] Zhixuan Z, Jun W, Jipeng G, et al. Support Vector Machine Process Against Probabilistic Byzantine Attack for Cooperative Spectrum Sensing in CRNs[C]//Proceedings of the 2023 8th International Conference on Machine Learning Technologies. 2023: 269-276. [R2] Chen Z, Wu J, Bao J. Semi-supervised Learning-enabled Two-stage Framework for Cooperative Spectrum Sensing Against SSDF Attack[C]//2022 IEEE Wireless Communications and Networking Conference (WCNC). IEEE, 2022: 554-559. [R3] Cheng Z, Song T, Zhang J, et al. Self- organizing map-based scheme against probabilistic SSDF attack in cognitive radio networks[C]//2017 9th International conference on wireless communications and signal processing (WCSP). IEEE, 2017: 1-6. [R4] Zhu H, Song T, Wu J, et al. Cooperative spectrum sensing algorithm based on support vector machine against SSDF attack[C]//2018 IEEE international conference on communications workshops (ICC workshops). IEEE, 2018: 1-6. |
Reference information provided by the reviewer is added, discussed, and expanded. The information is expanded in the final discussion section prior to the conclusions and the respective references are added to the list. |
|
Reviewer 3 |
6. The discussion section should be presented in categories. |
The discussion section is categorized according to the reviewer's instructions. |
We look forward to your comments and hope to hear from you soon.
Thank you very much
_
The authors
Reviewer 4 Report
Comments and Suggestions for Authors
If the editors believe this is within the scope of Sensors, I think that this work can be published in sensors, but before that it is necessary to eliminate a number of comments, some of which seem serious to me:
1. A search in the publication databases allowed the authors to find not such a large number of works. In my opinion, this happened because the search criteria were not set optimally. For example, the term 'machine learning' is not always used by professionals in this field; sometimes instead the title may contain: neural network, artificial intelligence, self-educated system, etc. In addition, abbreviations are often used in titles: CNN, DLN, ML, GLM and others. As for optical sensors, there can also be plenty of variations: optical fiber sensor, fiber optic sensing, optical fiber displacement sensor, optical fiber temperature sensor, etc. Abbreviations are also used quite often: DAS, OBR, OFDR f-OTDR, BOCDA. ..
2. In fact, all scientific works on optical sensors can be divided into two large categories: methods for developing fiber-optic sensors and methods for using fiber-optic sensors. Was this difference taken into account in the analysis?
3. It would be great if the authors made the captions for the pictures self-sufficient. For example, Figure 5 shows a graph on two axes: the number of publications and the number of citations. The graph should be enough to understand the time period for which the data is provided and other details. The same goes for the rest of the graphs.
4. In Figure 6, 2016 is omitted. I also ask the authors to explain why this type of diagram was chosen (circle).
5. What is the meaning of lines 93-104 highlighted in red?
6. In my opinion, the genres of reviews are unsuccessfully mixed: first, the authors discuss publication indicators, then suddenly they go into the scientific and technical details of a particular manuscript. I do not insist on a clear separation, since my judgment is subjective, but if possible, it would be great to “smooth out” the transitions between the technical and bibliographic parts, or vice versa, to separate them.
7. I would ask the authors to carefully proofread the manuscript for typos, extra characters, etc.
8. Table 1 is at the end of the section, I would suggest moving it higher, right after the first mention.
Minor editing of English language is required
Author Response
February 25th, 2024
Dear
Sensors
Kind regards
In accordance with the suggestions of the reviewers in our article “Machine Learning Applications in optical fiber sensing: A Research agenda”, the following changes were made, properly marked with red letters in the article:
|
Reviewer |
Comment |
Answer |
|
Reviewer 4 |
6. The discussion section should be presented in categories. |
The discussion section is categorized according to the reviewer's instructions. |
|
Reviewer 4 |
1. A search in the publication databases allowed the authors to find not such a large number of works. In my opinion, this happened because the search criteria were not set optimally. For example, the term 'machine learning' is not always used by professionals in this field; sometimes instead the title may contain: neural network, artificial intelligence, self-educated system, etc. In addition, abbreviations are often used in titles: CNN, DLN, ML, GLM and others. As for optical sensors, there can also be plenty of variations: optical fiber sensor, fiber optic sensing, optical fiber displacement sensor, optical fiber temperature sensor, etc. Abbreviations are also used quite often: DAS, OBR, OFDR f-OTDR, BOCDA. .. |
We add this information in a new subsection of the discussion titled “Limitations,” so that future researchers are careful when interpreting the data. |
|
Reviewer 4 |
2. In fact, all scientific works on optical sensors can be divided into two large categories: methods for developing fiber-optic sensors and methods for using fiber-optic sensors. Was this difference taken into account in the analysis? |
The difference was not taken into account in the analysis, which is why another paragraph of limitations is added, so that future research can overcome this limitation contained in the study. |
|
Reviewer 4 |
3. It would be great if the authors made the captions for the pictures self-sufficient. For example, Figure 5 shows a graph on two axes: the number of publications and the number of citations. The graph should be enough to understand the time period for which the data is provided and other details. The same goes for the rest of the graphs. |
The description of the images in each caption is improved and expanded according to the reviewer's suggestion for each of the images. |
|
Reviewer 4 |
4. In Figure 6, 2016 is omitted. I also ask the authors to explain why this type of diagram was chosen (circle). |
The necessary explanations are added at the end of the analysis of this figure in the discussion section |
|
Reviewer 4 |
5. What is the meaning of lines 93-104 highlighted in red? |
This was an editing error prior to submission, it is corrected |
|
Reviewer 4 |
6. In my opinion, the genres of reviews are unsuccessfully mixed: first, the authors discuss publication indicators, then suddenly they go into the scientific and technical details of a particular manuscript. I do not insist on a clear separation, since my judgment is subjective, but if possible, it would be great to “smooth out” the transitions between the technical and bibliographic parts, or vice versa, to separate them. |
Some transition phrases are added to indicate to readers that some technical aspects will be detailed according to the indicator being treated. |
|
Reviewer 4 |
7. I would ask the authors to carefully proofread the manuscript for typos, extra characters, etc. |
A grammar and spelling review process is carried out throughout the article |
|
Reviewer 4 |
8. Table 1 is at the end of the section, I would suggest moving it higher, right after the first mention. |
Table 1 is moved to the section immediately preceding the research agenda. |
We look forward to your comments and hope to hear from you soon.
Thank you very much
_
The authors
Round 2
Reviewer 2 Report
Comments and Suggestions for Authors
I advice to accept the paper in this new form.
Reviewer 3 Report
Comments and Suggestions for Authors
I have no further questions about this paper.
Reviewer 4 Report
Comments and Suggestions for Authors
Thank you for the updated paper, I propose to accept it in present form